# Arab Women Adherence to the Mediterranean Diet and Insomnia

**DOI:** 10.3390/medicina58010017

**Published:** 2021-12-23

**Authors:** Ayah T. Zaidalkilani, Omar A. Alhaj, Mohamed F. Serag El-Dine, Feten Fekih-Romdhane, Maha M. AlRasheed, Haitham A. Jahrami, Nicola L. Bragazzi

**Affiliations:** 1Department of Nutrition, Faculty of Pharmacy and Medical Sciences, University of Petra, Amman 11196, Jordan; ayah.zaidalkilani@uop.edu.jo; 2Department of Nutrition and Food Science, Faculty of Home Economics, Menofia University, Shibin el Kom 11925, Menofia Governorate, Egypt; serag_m1956@yahoo.com; 3Psychiatry Department “Ibn Omrane”, Razi Hospital, Manouba 2010, Tunisia; feten.fekih@gmail.com; 4Faculty of Medicine of Tunis, Tunis El Manar University, Tunis 1068, Tunisia; 5Clinical Pharmacy Department, College of Pharmacy, King Saud University, Riyadh 11451, Saudi Arabia; mahalrasheed@ksu.edu.sa; 6Ministry of Health, Manama 410, Bahrain; hjahrami@health.gov.bh; 7Department of Psychiatry, College of Medicine and Medical Sciences, Arabian Gulf University, Manama 323, Bahrain; 8Laboratory for Industrial and Applied Mathematics, Departments and Statistics, York University, Toronto, ON M3J 1P3, Canada

**Keywords:** Mediterranean diet, insomnia, Arabic-speaking, women, female, dietary patterns

## Abstract

*Background and objective:* Sleeping difficulties affect the overall health, nutrition, and wellbeing. The Mediterranean diet has proven effective in improving the quality of life and overall health of people of all ages. Therefore, this study aimed to determine whether adherence to the Mediterranean diet (MD) is linked to reduced insomnia in Arabic-speaking female adults in Jordan. *Materials and Methods:* A self-administered, cross-sectional survey was used to detect the relationship between MD and sleep quality in Arabic-speaking female adults. Data were collected from 917 Arabic-speaking female participants of 14 Arab nationalities in Jordan between March and May 2021 via social media. All participants answered the whole questionnaire, including questions on sociodemographic aspects, Mediterranean diet adherence, sleeping habits measured with the Athens Insomnia Scale (AIS), and lifestyle components such as smoking and dietary patterns. *Results:* The mean age of the 917 Arabic-speaking female participants was 36 ± 10 years. Most participants were unemployed (85%) single females (64%) with an undergraduate degree (74%). Most of them (86%) were non-smokers. More than half of the participants were Jordanians (57%). The BMI was normal for 52% of the participants, whereas 26% of them were overweight, and 12% were obese. One-way ANCOVA showed a statistically significant difference between MD adherence score categories and AIS, F (2, 914) = 3.36, *p* = 0.015. Among the MD adherence score categories, we found that between groups, MD scores above or equal to 10 were associated with a statistically significant difference in AIS. Cohen’s value was calculated for the three MD score categories and indicated a ‘small’ effect size association between all adherence scores of the MD categories and AIS. *Conclusions:* In conclusion, our findings provide preliminary evidence that participants’ adherence to the MD was significantly associated with better sleep and reduced insomnia symptoms, highlighting the need for further research.

## 1. Introduction

Insomnia is a sleep difficulty characterized by early awakening, difficulty in falling asleep or keeping a regular sleep, and insufficient restorative sleep [1]. The prevalence of insomnia is increasing, affecting 37.2% of the American population, 15.3% of Turkish adults [2], 6–36% of adolescents in Kuwait, 60% of Saudi adults, and 27% of Qatar soccer players [2]. Insomnia also affects our daily life as one of the primary causes of cognitive difficulties [3], memory deficits [3,4,5,6], and reduction in daily productivity, leading to chronic sleep deprivation [3,6,7,8]. Insomnia is linked to food intake and diet quality by increasing food sensitivity and altering appetite hormones [9]. Research shows that sleep disorders may have a more severe impact on women’s health compared to men’s [3]. Other studies showed that diet might influence sleep metrics, meaning that nutrition significantly affects sleep quality [9,10,11]. For example, preliminary studies showed that a high-carbohydrate diet reduces sleep latency onset, whereas a high protein diet enhances sleep. Conversely, too much fat in the diet may affect overall sleep time. In line with these studies, limited trials have shown that the intake of foods with high levels of tryptophan, melatonin, and other different nutrients can affect sleep [12]. Therefore, the MD has raised a particular interest for its potential to improve sleep health [10]. Notably, for sleep, the MD encourages the intake of high-nutrient food groups and other foods that are high in protein derived from plants and unsaturated fatty acids [13], which promotes sleep [14].

The Mediterranean diet has grown in popularity over the last few decades due to its delectable flavor and substantial evidence supporting its health benefits [15]. The MD has no specific definition; its properties were explored by Ancel Keys in the 1960s, who studied the traditional diet of Southern Italians, as certain peculiarities characterize it. In the MD, the major fat source is olive oil and it includes moderate quantities of dairy (yoghurt and cheese), low amounts of red meat, a moderate fish amount, moderate amounts of wine with the meals, a high intake of fruits and vegetables, and an active lifestyle [6,7,8]. Although diet varies in the various Mediterranean areas due to societal, religious, and economic circumstances, these are considered variants of the MD [6,7,8]. Not surprisingly, the MD improves sleep length and sleep difficulty indicators [8,9,10,11,12,13,14,15]. Therefore, the MD is considered a healthy diet based on reduced total calories and added sugars and increased intake of fruits, vegetables, raw nuts, and unsaturated fats [7,8].

Adherence to the MD was related to enhanced sleep and reduced insomnia symptoms [4,5,6]. Accordingly, it is critical to assess MD adherence using precise assessment measures such as dietary scores based on food consumption patterns and evaluation of to the recommended intake [16]. Adherence to the MD has also been linked to several favorable cardiovascular and metabolic outcomes [17,18], lowering the risk of heart illnesses [19,20], diabetes [21], metabolic disorders [22,23], and non-alcoholic fatty liver disease [24,25]. These positive outcomes are achieved through various mechanisms, most of which involve antioxidants and healthy dietary fats, which may improve insulin resistance, decrease infection risk, and ameliorate coronary vascular insufficiency [26]. Recent evidence suggests that following an MD may also be beneficial to mental health and short-term cognitive effects, such as neurological disorders, stroke, cognitive decline, depression, and dementia [27,28,29,30,31,32]. In addition, recent studies indicate a correlation between MD adherence and sleep duration in adults [33,34,35].

Additionally, former cross-sectional studies have shown correlations between nutrient-dense foods found in the MD, specifically, fruits, vegetables, and unsaturated fatty acids, and sleep measures [4,36]. The current study sought to determine whether adherence to the MD predicts habitual sleep patterns in Arabic-speaking female adults. As such, we aimed to investigate if following the MD or its key components was related to improved sleep in Arabic-speaking female adults.

## 2. Materials and Methods

### 2.1. Study Design

This study was designed as a self-administered, cross-sectional, multinational, and observational research to examine the adherence of Arabic-speaking females to the MD and its association with insomnia symptoms. A total of 917 female adults were surveyed using a convenient sample and a validated questionnaire whose answers were evaluated using the MD score tool (MD adherence score) and the Athens Insomnia Scale (AIS) [37]. In our study, we adopted Trichopoulou et al. and PREDIMED (Prevención con Dieta Mediterránea) methodology to assess adherence to the traditional MD [38,39]. Beneficial components (vegetables, legumes, fruits, nuts, cereal, and fish) were given a rating of 0 or 1. Consumption of harmful components was given a score of 1 if less than the median was consumed (meat, poultry, and dairy products, which are seldom non-fat or low-fat). Accordingly, MD adherence scores varied from <6 (low adherence of the MD) to ≥10 (high MD adherence); a score between 6 to 9 indicated moderate adherence to the MD.

The sample size was estimated for our survey to be around 385 participants, assuming a confidence level of 95% (95% CI) that the real value was within 5.0% (alpha level), and type II error was about 20% of the surveyed values of the female population in the Middle East and North Africa (223,874,467) according to the latest World Bank statistics [3]. In the final analysis, we examined 917 individuals to raise the statistical power to 90 %.

This study was conducted in Jordan, following the “Method for Observational Studies in Nutritional Epidemiology” (STROBE-nut) guidelines to enhance the quality of the study design and report [40].

### 2.2. Study Setting and Participants

The study included a total of 917 Arabic-speaking female participants of 14 Arab nationalities (Syria, Palestine, Kingdom of Saudi Arabia, Egypt, Yamen, Lebanon, Algeria, Sudan, Morocco, Libya, Iraq, Kuwait, Bahrain, Western Sahara) living in Jordan. The data were collected consecutively between March and May 2021 using convenient self-selection adult sampling. The participants response rate to the online survey was 89.5%, and all questions were answered. The participants were chosen from the general population based on the following inclusion criteria: (1) a female adult over the age of 18, (2) capable of writing and speaking Arabic, (3) willing to give information and volunteer to participate in the study. Participants who did not meet the inclusion criteria were excluded. Additionally, participants who did not consent or wished to withdraw were excluded from the study.

Data were collected through an online survey via instant electronic communication using Google forms. The survey was mainly sent through advertisements on social media like WhatsApp, BlackBerry Messenger, Viber, Signal, Line, Facebook, Twitter, Instagram. Other recruitment methods were also used using word of mouth, work relationships, and authors’ connections. Moreover, participants were requested to share the link to the questionnaire with additional participants who fit the inclusion criteria via social media. All questions in our survey were translated into Arabic and made as straightforward as possible. Participants who were willing to write about their dietary consumption, symptoms, health, and lifestyle habits were allowed to complete the questionnaire in their spare time. All data were saved in a secure Google Drive, accessible only to the principal investigator and coded for the research team.

Weight and height were obtained from the self-reported questionnaire, and the body mass index (BMI) was generated accordingly.

### 2.3. Ethical Approval

This study received ethical approval on the 14th of January 2021 from the Faculty of Pharmacy and Medical Sciences, University of Petra, Amman, Jordan (UOP/REC: 1H/1/2021). It was carried out in conformity with the Declaration of Helsinki of 1964. By signing a consent form, all participants agreed to participate in this study. Participants were able to leave the study at any time without obligation or notice.

### 2.4. Instrument

The survey comprised three sections of questions. Initially, the participants reported sociodemographic information, anthropometric measurements, educational level, smoking habits, country of origin, and general health status. Then, the participants completed the MD score tool, which comprises of 14 questions, answered with yes or no. The MD includes high levels of protective foods, such as fruits, vegetables, legumes, whole grains, fish, and olive oil, with moderate red meat and alcohol consumption. The MD emphasizes the consumption of whole or minimally processed foods and a low consumption of sugar-sweetened beverages, refined grain products, and processed or energy-dense foods [6,7,8,13,14,15]. A high adherence score of the MD is related to a reduced cardiovascular disease (CVD) risk and all-cause mortality.

Finally, in the third section of the survey, the Athens Insomnia Scale (AIS) was used to diagnose insomnia cases in adults over the previous month [37]. This test has an excellent internal consistency (Cronbach alpha = 0.90). AIS is a self-assessment psychometric instrument based on an eight-item test: the first five items are about sleep induction, night-time awakenings, final awakening, total sleep duration, and sleep quality; the last three items are about well-being, functioning capacity, and daytime sleepiness. Each item is evaluated with a four-point Likert scale. The Arabic validated AIS version [41] was used, which has a high Cronbach’s alpha of about 0.83.

### 2.5. Statistical Analysis

As part of the data analysis, histograms and box plots were used to identify potential outliers and test the normality of the study variables using the Shapiro–Wilk test. To summarize the data, descriptive statistics were used, including means and standard deviations (for continuous variables) and absolute and relative frequencies (for categorical variables). An insomnia diagnosis can be made based on a score of ≥6 on the AIS. We examined adjusted differences between participants based on insomnia status with the lowest adherence to the Mediterranean diet (score < 6, reference category) and those with moderate and high degrees of adherence (6–9, or ≥10). The analysis of covariance (ANCOVA) was performed to detect if there were any statistically significant differences in the means of three or more unrelated groups based on MD categories regarding the AIS score.

We also examined the effect size using Cohen’s d. The effect size was estimated via Cohen’s d and was interpreted as follows: 0.20 as small, 0.50 as moderate, ≥0.80 as large. After the omnibus F test, post hoc tests were performed to locate those specific differences. Post hoc comparisons are based on estimated marginal means using Tukey’s correction. We used two-tailed tests, and results with *p* values less than 0.05 were considered statistically significant. All statistical analyses were performed using R for statistical computing 4.0.3.

## 3. Results

### 3.1. Study Sample Characteristics

As shown in Table 1, the mean age of the 917 Arabic-speaking female participants was 36 ± 10 years. The average weight of the participants was 63.4 ± 14 kg, and the average height of the women who participated in the study was 161.5 ± 6 cm. Most participants were unemployed (85%) single females (64%) with an undergraduate degree (74%). Almost 86% were non-smokers. More than half of the participants were Jordanians (57%). BMI was normal for 52% of the Arabic-speaking female participants, whereas 26% were overweight, and 12% were obese.

### 3.2. Mediterranean Diet Adherence Score

As shown in Table 2, One-third of the Arabic-speaking female participants (31%) scored less than 6 for the adherence to the MD, (MD Score < 6). Moderate adherence to the MD was shown by 56% of the study participants (MD Score 6–9), whereas high adherence to the MD (scores ≥ 10) was seen for 13% of the study sample.

### 3.3. Sleep Difficulty

As shown in Table 3, almost half of the participants (52%) reported to sleep between 6 and 8 h a day. At the same time, 43% of them needed an hour to go to sleep. Most study participants reported either to not wake up during the night (41%) or to wake up once (41%). About 88.9% of the participants reported not to wake up earlier or a little earlier than the desired time. Furthermore, the participants reported a satisfactory (47%) or slightly satisfactory (36%) sleep quality, while their degree of comfort during sleeping was satisfactory (48%).

### 3.4. Association between Adherence to the MD and AIS

Results from one-way ANCOVA showed a statistically significant difference between MD adherence score categories and AIS status, F (2, 914) = 4.03, *p* = 0.018. Therefore, we rejected the null hypothesis H_0_, which stated no significant difference between MD scores and AIS status. The alternative hypothesis (H_1_) stated that the existence of a significant difference between MD scores and AIS status.

### 3.5. Post Hoc Analysis for Multiple Comparisons

Among the MD adherence score categories, we found that between groups, MD adherence scores of less than 6 and above 10 had no statistically significant difference. Additionally, there were no statistically significant differences in the MD adherence score for MD score 6–9. Participants who scored >10 alone had better AIS, *p*-value = 0.015. Cohen’s value calculated for the three MD adherence scores indicated a ‘small’ effect size association between all MD adherence scores and the AIS status, as shown below.

### 3.6. Estimated Marginal Means of MD Categories

The model estimated marginal means and standard errors of AIS at the factor level of the different MD adherence categories. AIS was better among participants with an MD adherence score ≥ 10, with a mean increase of 6.02, 95% CI (5.28, 6.76) and was statistically significant (*p* = 0.015), but no other group differences were statistically significant.

## 4. Discussion

In our study, 917 Arabic-speaking female adults of 14 Arab nationalities who live in Jordan were included. Half of the participants had a normal BMI (18.5–24.9 kg/m^2^). Nonetheless, the adherence to the MD was moderate for 56% of the sample. AIS was better for participants with an MD adherence score ≥ 10, with a mean increase of 6.02, 95% CI (5.28, 6.76) which was statistically significant (*p* = 0.015), but no other group differences were statistically significant.

Recent research examined the connection between sleep quality and adherence to the MD of Middle Eastern Arab individuals [18,35]. Overall, sleep quality was significantly linked to a healthy diet [42,43,44]. Few studies have previously examined the connection between MD adherence and sleep parameters in adults [42,43]. A research study enrolled 1500 Spanish seniors for almost 3 years, measuring sleep duration and sleep quality markers [42]. According to the researchers, people who followed the MD had a reduced chance of changing their sleep routine and had better sleep quality [42]. Another research found that adhering to the MD helped to improve sleep issues, including insomnia symptoms [43]. Several studies investigated the general connection between sleep length and diet quality [44,45,46,47,48], while others addressed sleep patterns and eating behaviors such as skipping meals, eating too little, eating meals at irregular times, and snacking between meals [6,7,8,9,10,45,46,47,48]. Experimental research has proven that high-quality diets enhance sleep duration, while sleep deprivation has been shown to increase the desire for high-calorie meals [6,10,43,44,45,46,47,48].

The MD tends to include high amounts of fruits, vegetables, seafood, whole grains, and olive oil, while minimizing meat, dairy, and alcohol [1]. It has also been suggested that better sleep and better mental health are linked to the MD [45,46]. Bioactive substances like antioxidants and anti-inflammatory components of the MD may serve as neurological protective agents, decreasing oxidative damage and brain ischemia [40]. Indeed, during sleep deprivation, elevated oxidative processes have been observed in many human internal organs like the liver, the brain, and the heart. Neurological inflammation was suggested to have a direct relationship to poor sleep [37,41,44]. Additional data also evidenced the role of some vitamins, specifically vitamin D and vitamin C, which may have a direct relationship to better sleep for either sleep length or sleep quality [47]. In addition, poor mental health is correlated with low levels of polyunsaturated fatty acids (PUFAs) in the diet [47]. Other studies emphasized the role of exercise, sleep, and dietary habits in reducing stress, anxiety, and depression, as well as cognitive dysfunction [2,3,48,49]. Many components of the MD may improve the structural and functional brain plasticity whose deficit is associated with Alzheimer’s disease [48,49].

Overweight people may develop sleep apnea, which leads to poor sleep quality and sleep deprivation, which is proven to be positively associated with obesity [49,50,51]. Some new research has shown that sleep apnea may cause blood vessel problems, affecting cognition and leading to death and illness [52].

### Strength and Limitations

To our knowledge, no previous study was conducted to assess the link between Mediterranean diet adherence and insomnia among Arabic-speaking women in Jordan. We collected data from 917 female participants of Jordanian nationality as well as of 14 Arab nationalities using a validated MD adherence score [39,40] and AIS questionnaire [1]. Our findings showed that high MD adherence was linked significantly to improved sleep. Insomnia affects women more than men, also in the 21st century. Some women are simultaneously caring for children and aging parents. Women may also work outside the house while caring for others. These activities may affect women’s sleep. In addition, hormonal changes in women’s life phases may impact sleep quality. For example, women with premenstrual dysphoric disorder (PMDD), a severe type of PMS, are more prone to sleeplessness. During pregnancy, women may experience severe leg or stomach pain. Menopausal women often suffer from sleeplessness owing to hot flashes and excessive perspiration. Women are also more prone to mood disorders like worry and sadness, making sleep issues more prevalent. Finally, women are more likely to be primary caregivers. Therefore, conducting this study on women is one of the strengths of this work, due to the lack of specific studies in the Middle East region.

The multi-ethnic features of the examined population represent a snapshot of each Middle Eastern country. This strengthens our study, capturing diverse characteristics of dietary intake. The limitation of our study is related to its cross-sectional design and the fact that it was based on a self-reported questionnaire on MD and insomnia, subjected to memory and social desirability biases. These instruments were extensively researched, and the methodologies employed in this study are nearly the same as those used in previous studies. Nonetheless, this study was performed in Jordan and as such it is not representative of the entire geographical area of the Middle East. In addition, the study participants were females, which further limits the generalizability of the results. The low causality of this type of study and the use of bivariate descriptive analysis also limit the generalizability of our findings.

## 5. Conclusions

In conclusion, our findings provide preliminary evidence that participants’ adherence to the MD was significantly associated with better sleep. We also wish to emphasize the importance of good nutrition, high adherence to the MD, and a healthy lifestyle in reducing insomnia symptoms amongst Arabic-speaking female adults. It is worth mentioning that other factors may influence insomnia in females, such as advanced age and hormonal changes regarding sex steroids, which were not studied in our research. Therefore, further research is needed to include the role of female sex steroids in insomnia among Arabic-speaking female adults.

## Figures and Tables

**Table 1 medicina-58-00017-t001:** Sociodemographic characteristic of the 917 Arabic-speaking female participants.

Study Characteristics	(N = 917)
Age (years)	36.32 (10.37)
Weight (Kg)	63.36 (13.95)
Height (cm)	161.49 (6.05)
BMI (kg/m^2^)	24.25 (4.94)
Relationship Status	
Single	580 (63%)
Married/Cohabit	318 (35%)
Divorced	13 (1%)
Widowed	6 (1%)
Smoking status	
Current smoker	129 (14%)
Not a smoker	788 (86%)
Level of education (years)	
Secondary level (9–11 years of education)	194 (21%)
Undergraduate level	376 (74%)
Postgraduate level	47 (5%)
Current Occupational status	
Not currently employed ^1^	779 (85%)
Employed	138 (15%)
BMI category ^2^	
Underweight (<18.5 kg/m^2^)	85 (9%)
Normal (18.5–24.9 kg/m^2^)	477 (52%)
Overweight (25–29.9 kg/m^2^)	241 (26%)
Obese (>30 kg/m^2^)	114 (12%)

Continuous variables are summarized as mean (standard deviation). Categorical variables are summarized as n (%), where n is the number of available cases (percentage of available cases); missing values were taken into consideration by calculating the available cases rather than all cases. ^1^: Not currently employed includes students, housewives, and never employed. ^2^: BMI (Body Mass Index) according to the WHO classification.

**Table 2 medicina-58-00017-t002:** Mediterranean diet adherence score for 917 Arabic-speaking female participants.

MD Score	Responses (%)
<6	284 (31%)
6–9	518 (56%)
≥10	115 (13%)

MD Scores: Mediterranean diet adherence score. Scores < 6 were considered as low adherence; between 6 and 9 as moderate adherence, ≥10 as high adherence to the MD. Categorical variables are summarized as n (%), where n refers to the responses of the available cases (percentage of available cases); the missing values were taken into consideration by evaluating the available cases rather than all cases.

**Table 3 medicina-58-00017-t003:** Sleep difficulty assessment in the 917 female participants.

Questions	Responses (%)
How many hours do you sleep per day?	
Less than 4 h (<4 h/day)	21 (2%)
More than 8 h (>8 h/day)	167 (18%)
Between 4–6 h a day	252 (27%)
Between 6–8 h a day	477 (52%)
The beginning of sleep: the time you need to go to sleep, after turning off the lights	
No time needed to fall asleep	218 (24%)
Slightly late	390 (43%)
Very late	209 (23%)
I stay awake and cannot sleep	100 (11%)
Waking up during the night (how many times?)	
No problem	376 (41%)
Simple problem	387 (42%)
A problem worth paying attention to	108 (12%)
A serious problem, or I could not sleep at all.	46 (5%)
Waking up earlier than desired	
I do not wake up earlier than desired	423 (46%)
Wake up a little earlier than desired	393 (43%)
I wake up clearly earlier than desired	101 (11%)
Total amount of sleep	
Sufficient	420 (46%)
Slightly insufficient	355 (39%)
Insufficient	118 (13%)
Very insufficient, or I could not sleep at all	24 (3%)
Overall sleep quality, regardless of the amount of sleep or the number of sleeping hours	
Satisfactory	430 (47%)
Slightly satisfactory	332 (36%)
Slightly unsatisfactory	129 (14%)
Very unsatisfactory	26 (3%)
Feeling comfortable during sleep	
Satisfactory	439 (48%)
Slightly satisfactory	333 (36%)
Slightly unsatisfactory	79 (9%)
Very unsatisfactory	66 (7%)
Physical and psychological functional performance during the day, i.e., your ability to perform your physical and psychological tasks during the day	
Normal	428 (47%)
Slightly deficient	305 (33%)
Deficient	140 (15 %)
Very clearly deficient	44 (5%)
Drowsiness during the day—Feeling sleepy	
None	120 (13%)
Slightly sleepy	543 (59%)
Sleepy	202 (22%)
Very sleepy	52 (6%)
AIS	
Healthy	378 (41%)
Insomnia	539 (59%)

AIS: Athens Insomnia Scale. Categorical variables are summarized as n (%), where n refers to the responses of the available cases (percentage of available cases). Missing values were taken into consideration by evaluating the available cases rather than all cases.

## Data Availability

The data are available from the authors upon reasonable request.

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
