# Peer review of "Arab Women Adherence to the Mediterranean Diet and Insomnia"

_medicina, 2021, doi:10.3390/medicina58010017_

Round 1

Reviewer 1 Report

Dear authors,

This paper deals with a very interesting topic of study. MD has extensive antioxidant and nutritional properties that may influence sleep quality. In fact, there are several studies of great relevance, such as the PREDIMED-PLUS study that has studied this association, although not in the Jordanian population. However, it is a bit difficult to read. Sometimes because of the way it is written and sometimes because of the density of the information presented. A second reading of the work is necessary in order to eliminate information that does not add much for the reader. This is particularly true in the results section and in the abstract. The methodology section should also be revised and completed with some of the suggestions I propose. Another important aspect to be addressed is the lack of references in some sections.

Abstract

The abstract is long but provides little information. In my opinion it would be more useful to focus the information presented on insomnia and adherence to MD.

  • Lines 24-26: Focus information on questionnaires used to assess adherence to MD and sleep quality.
  • Line 26: AIS or ISI?
  • Methods: If possible, add information on statistical analysis.
  • Lines 27-28: “most of the participants (74%) were educated, single, holding an undergraduate degree, not obese, and not smokers.” This information is not clear, did 74% of the participants have all these characteristics at the same time? In my opinion, all this information can be exchanged for % on sex and average age.
  • Lines 28-32: Is it key to present this information? If it is, it is necessary to revise and try to draft in a more readable and concrete way.
  • Lines 33-39: In my opinion, these are the key results in line with your study objective. However, they are difficult to understand because you have not previously explained what the categories of adherence to the MD mean. It is also necessary to explain whether a higher AIS score means greater or lesser severity.
  • Line 31: What MD adherence? High, medium, low? Please, clarify.

Introduction

Line 53: Please, add some reference.

Line 54: “This is linked…” What means “This” insomnia or the negative effects of the insomnia? Or perhaps you mean that insomnia can also be influenced by food intake? Please, clarify.

Lines 56-57: Please, add some reference.

Line 61: This is the first time you refer to the “MD” in the introduction, please describe it.

Line 65: Use MD.

Methods:

Line 106-107: The insomnia scale here is not the same as in the abstract.

Line 122: Repeated information.

Line 137: It would be very useful to add here information on the scores and categories of the MD adherence and insomnia scales. Also, you should include the reference of the MD tool, you may find it here: DOI: 10.1016/j.clnu.2021.06.030 In general, a more in-depth description of the study variables would be very useful.

Line 159-160: How did you represent non-parametric continuous variables? Please, add this information.

Line 165: ANCOVA or ANOVA?

 Results

Line 178: One-third of 1025 participants? Please, revise this “n=54”.

Lines 181-184: I suggest that you include this information in the table and remove it from the text. In my opinion, it is sufficient to indicate that the majority of participants were from Jordan.

Line 190: This sentence is difficult to read, I suggest you change it to:…almost half of the study population (48%) reported consuming less than three meals a day, while 36% consumed exactly three meals a day, and 15% consumed more than three meals a day.

Line 192: Why do you put 59% in brackets? Please check the use of brackets throughout the results.

Line 192: According to MDPI formatting standards: numbers 0–9 should be written as words unless they are a measurement, i.e., they are accompanied by a unit. For example: five trees - 5 m from the tree. If a sentence starts with a number, the number should always be written out in full, however it is often better to reword the sentence. Thus, write “4” as words, check this throughout the manuscript.

Line 215-216: “…while their satisfaction during sleep was 48.8%.” Satisfaction or comfort? Please, clarify.

Table 3: Add a table footer.

Subsection 3.4: From the information you provide, it is clear that there is a difference between the categories of adherence to the MD and the ISA. But how do we know what % of people with insomnia are in each MD adherence category? I think you should add more information to help the reader interpret the results.

Line 233: Possible mistake: Table 5.

Discussion:

Line 254-255: “Overall, sleep difficulty was significantly linked with a healthy diet.”  Is this correct?

Lines 271-275: Please, add some reference.

Lines 312-313: In my opinion, this statement cannot be derived from your results.

Limitations: It would be appropriate to add the possible limitations of a non-random, convenience and voluntary sample. In this respect, it is worth noting that 90% of the participants were women, which further limits the generalisability of the results. It would also be convenient to add to the limitations the low causality of this type of study and bivariate descriptive analysis.

Conclusions: After reading the conclusions, it is still not clear to me whether increased adherence to MD is related to better sleep quality, reduced insomnia or increased insomnia. In fact, in the first paragraph of the discussion it is implied that MD is associated with more insomnia. Please review the text and try to specify and clarify the key information in your paper.

Thank you for your work, I hope you find my comments useful.

Author Response

Reviewer One Comments and feedback

Dear reviewer,

Thank you for the amazing feedback and your time. Your efforts are much appreciated. You will find your comments with the response in a red color font.

Abstract

The abstract is long but provides little information. In my opinion it would be more useful to focus the information presented on insomnia and adherence to MD.

Response:

“Abstract:

Background and objective: Sleeping difficulty affects the overall health, nutrition, and wellbeing. The Mediterranean diet has proven its effect on improving the quality of life and overall health outcomes among people of all ages. Therefore, this study aims to determine whether adherence to a Mediterranean diet (MD) predicts reduced insomnia in Arabic-speaking female adults in from fourteen Arab countries.

Methods: A self-administered, cross-sectional survey was used to detect the relationship between MD and sleep quality in Arabic-speaking female adults. The data was collected from 917 Arabic-speaking female participants in Jordan with fourteen Arab nationalities between March and May 2021 via social media. All participants have answered the whole questionnaire, including sociodemographic questions, Mediterranean diet score tool, Athens Insomnia Scale (AIS), and lifestyle components including smoking and dietary patterns.

Results: Of the 917 Arabic-speaking female participants, the mean age was 36 ± 10 years. Most participants were unemployed (85%) single females (64%) and were holding an undergraduate degree (74%). Almost (86%) were non-smokers. More than half of the participants were Jordanians (57%). BMI was normal amongst (52%) of the Arabic-speaking female participants, (26%) were overweight, and (12%) were obese. One-way ANCOVA showed a statistically significant difference between MD adherence score categories and AIS, F (2, 914) = 3.36, p = 0.015. Among the MD adherence score categories, we found that between groups, MD scores of above or equal to ten had a statistically significant difference in AIS. Cohen's value calculated for the three MD score categories and indicated a 'small' effect size association between all adherence scores of the MD categories and AIS.

Conclusion: In conclusion, findings provide preliminary evidence that participants’ adherence to the MD was significantly associated with better sleep, reduced insomnia symptoms and highlights the need for further research.”

  • Lines 24-26: Focus information on questionnaires used to assess adherence to MD and sleep quality.

Response: the information was added, and the content was modified.

  • Line 26: AIS or ISI?

Response: AIS

  • Methods: If possible, add information on statistical analysis.

Response: Information were added.

  • Lines 27-28: “most of the participants (74%) were educated, single, holding an undergraduate degree, not obese, and not smokers.” This information is not clear, did 74% of the participants have all these characteristics at the same time? In my opinion, all this information can be exchanged for % on sex and average age.

Response: the sentence was modified and rephrased.

A self-administered, cross-sectional survey was used to detect the relationship between MD and sleep quality in Arabic-speaking female adults. The data was collected from 917 Arabic-speaking female participants in Jordan with fourteen Arab nationalities between March and May 2021 via social media. All participants have answered the whole questionnaire, including sociodemographic questions, Mediterranean diet score tool, Athens Insomnia Scale (AIS), and lifestyle components including smoking and dietary patterns.”

  • Lines 28-32: Is it key to present this information? If it is, it is necessary to revise and try to draft in a more readable and concrete way.

Response: the sentence was removed.

  • Lines 33-39: In my opinion, these are the key results in line with your study objective. However, they are difficult to understand because you have not previously explained what the categories of adherence to the MD mean. It is also necessary to explain whether a higher AIS score means greater or lesser severity.

Response: we have added the following detailed explanation. The AIS is commonly used in medicine to assess insomnia [39,40]. It is assessed by evaluating eight parameters, the first five of which are connected to nocturnal sleep and the final three of which are associated to daytime dysfunction [39]. These are assessed on a scale of 0–3, and the sleep is eventually evaluated based on the sum of all aspects and presented as an individual's sleep result. Over time, AIS has proven to be a successful tool in sleep analysis, and it has been validated in several nations by testing it on local patients [39,40]. The AIS has a cut-off score of 6 to establish the diagnosis of insomnia; hence, a higher AIS score indicates a more severe form of insomnia [39,40].

  • Line 31: What MD adherence? High, medium, low? Please, clarify.

Response: The adherence was clarified.

Introduction

Line 53: Please, add some reference.

Response: references were added as advised.

Line 54: “This is linked…” What means “This” insomnia or the negative effects of the insomnia? Or perhaps you mean that insomnia can also be influenced by food intake? Please, clarify.

Response: the sentence was clarified

Lines 56-57: Please, add some references.

Response: references were added as advised.

Line 61: This is the first time you refer to the “MD” in the introduction, please describe it.

Response: the MD was described well in the text as:

"In the Mediterranean diet (MD), the major fat source is olive oil, with moderate quantities of dairy (yoghurt and cheese), low amounts of red meat and a moderate fish amount, moderate amounts of wine with meals, high intake of fruits and vegetables, and an active lifestyle [11–14]. Although the diets of the many Mediterranean areas vary due to societal, religious, and economic circumstances, it is thought that they are all variants of the MD diet [11–14]. "

Line 65: Use MD.

Response: we used MD

Methods:

Line 106-107: The insomnia scale here is not the same as in the abstract.

Response: we unified insomnia scale both in the abstract and the methods using the Athens Insomnia Scale (AIS).

Line 122: Repeated information.

Response: The repeated information was Deleted

Line 137: It would be very useful to add here information on the scores and categories of the MD adherence and insomnia scales. Also, you should include the reference of the MD tool, you may find it here: DOI: 10.1016/j.clnu.2021.06.030 In general, a more in-depth description of the study variables would be very useful.

Response: thank you for your suggestion and it was considered.

Line 159-160: How did you represent non-parametric continuous variables? Please, add this information.

Response: All data was normally distributed. No skewed variables were obtained from the data. Therefore, we did not use the non-parametric tests in the analysis.

Line 165: ANCOVA or ANOVA?

Response: The analysis of covariance (ANCOVA) was the test we used in our study

 Results

Line 178: One-third of 1025 participants? Please, revise this “n=54”.

Response: Corrected and revised.

“The majority of the participants were single females (64%) and (86%) were non-smokers.”

Lines 181-184: I suggest that you include this information in the table and remove it from the text. In my opinion, it is sufficient to indicate that the majority of participants were from Jordan.

Response: your comment was taken into consideration.

Line 190: This sentence is difficult to read, I suggest you change it to:…almost half of the study population (48%) reported consuming less than three meals a day, while 36% consumed exactly three meals a day, and 15% consumed more than three meals a day.

Response: the sentence was deleted due to the change in the sample (Females only).

Line 192: Why do you put 59% in brackets? Please check the use of brackets throughout the results.

Response: the use of brackets was checked throughout the results.

Line 192: According to MDPI formatting standards: numbers 0–9 should be written as words unless they are a measurement, i.e., they are accompanied by a unit. For example: five trees - 5 m from the tree. If a sentence starts with a number, the number should always be written out in full, however it is often better to reword the sentence. Thus, write “4” as words, check this throughout the manuscript.

Response: the guidelines were taken into consideration.

Line 215-216: “…while their satisfaction during sleep was 48.8%.” Satisfaction or comfort? Please, clarify.

Response: Comfort. It was corrected in the text.

Table 3: Add a table footer.

Response: footer was added to Table (3).

Subsection 3.4: From the information you provide, it is clear that there is a difference between the categories of adherence to the MD and the ISA. But how do we know what % of people with insomnia are in each MD adherence category? I think you should add more information to help the reader interpret the results.

Response: information was added in the paper

Line 233: Possible mistake: Table 5.

Response: corrected.

Discussion:

Line 254-255: “Overall, sleep difficulty was significantly linked with a healthy diet.”  Is this correct?

Response: this part of the discussion was modified.

Lines 271-275: Please, add some reference.

Response: references were added

Lines 312-313: In my opinion, this statement cannot be derived from your results.

Response: the statement was deleted.

Limitations: It would be appropriate to add the possible limitations of a non-random, convenience and voluntary sample. In this respect, it is worth noting that 90% of the participants were women, which further limits the generalisability of the results. It would also be convenient to add to the limitations the low causality of this type of study and bivariate descriptive analysis.

Response: we added the following:

“Nonetheless, this study was performed in Jordan, and it is not representative of the entire geographical area of the Middle East. In addition, the study participants were females, which further limits the generalizability of the results. The low causality of this type of study and the use of bivariate descriptive analysis played also a role in limiting the generalizability of our findings”

Conclusions: After reading the conclusions, it is still not clear to me whether increased adherence to MD is related to better sleep quality, reduced insomnia or increased insomnia. In fact, in the first paragraph of the discussion it is implied that MD is associated with more insomnia. Please review the text and try to specify and clarify the key information in your paper.

Response: the text was reviewed and modified.

Reviewer 2 Report

1.Introduction

The introduction should be improved by including references to the definition of the Mediterranean diet and the test used to measure adherence to it.

2. Materials and Methods

The countries in which the survey was conducted should be indicated.

The reference to the Mediterranean diet adherence questionnaire and whether a validated version for the Arab population is available should be indicated.

4. Discussion

The discussion should be improved as it is expected to compare the results obtained in the present study with what is in the literature and to elaborate on the reasons for similarities, differences and novel relationships, not a summary of the existing literature.

4.1. Strength and limitations

Limitations include the fact that most of the sample is from a single country, so it is unlikely to be representative of the entire geographical area under study.

5. Conclusions

The conclusions on obesity do not correspond with the data obtained in the study. Conclusions should focus on the results obtained.

Author Response

Reviewer Two Comments and feedback:

1. Introduction

The introduction should be improved by including references to the definition of the Mediterranean diet and the test used to measure adherence to it.

Response: references were added, and the text was modified and colored in red.

  1. Materials and Methods

The countries in which the survey was conducted should be indicated.

Response: the study was conducted in Jordan, and this was clarified in the paper.

The reference to the Mediterranean diet adherence questionnaire and whether a validated version for the Arab population is available should be indicated.

Response: the reference was added and the text was modified accordingly.

  1. Discussion

The discussion should be improved as it is expected to compare the results obtained in the present study with what is in the literature and to elaborate on the reasons for similarities, differences and novel relationships, not a summary of the existing literature.

Response: “In our study, 917 Arabic-speaking female adults with fourteen Arab nationalities who live in Jordan were included. Half of the participants were among the normal BMI (18.5 – 24.9 kg/m2). Nonetheless, the adherence of the MD diet was moderate amongst 56% of the sample. AIS was better among MD adherence score ≥10, with a mean increase of (6.02, 95% CI [5.28, 6.76]) and was statistically significant (p = 0.015), but no other group differences were statistically significant.”

4.1. Strength and limitations

Limitations include the fact that most of the sample is from a single country, so it is unlikely to be representative of the entire geographical area under study.

Response: we added the following statement:

“Nonetheless, this study was performed in Jordan, and it is not representative of the entire geographical area of the Middle East. In addition, the study participants were females, which further limits the generalizability of the results. The low causality of this type of study and the use of bivariate descriptive analysis played also a role in limiting the generalizability of our findings”

  1. Conclusions

The conclusions on obesity do not correspond with the data obtained in the study. Conclusions should focus on the results obtained.

Response: the conclusion was changed to best represent the results obtained.

In conclusion, findings provide preliminary evidence that participants’ adherence to the MD was significantly associated with better sleep. we also want to emphasize the importance of good nutrition, high adherence to the MD, and a healthy lifestyle in reducing insomnia symptoms amongst Arabic-speaking female adults. Worth mentioning that other factors may influence insomnia in females, such as, advanced age, hormonal changes of sex steroids, which were not studied in our research. Therefore, further research is needed to include the role of female sex steroids among Arabic-speaking female adults.

Reviewer 3 Report

Materials and methods:

  • How did you determine sample size? Based on specified inclusion criteria what was response rate?
  • How do you explain the difference in the number of participants by gender?
  • Ethical approval – Since it is multinational study, whether you should get ethical approval from all countries included in study? What happened to the participants who didn’t sign consent form?

Results:

  • You said that about 90% of participants were female. That is not about two-third as you wrote, because two-third is about 66%. It is similar with male. Also, you should tune percentages in text and in Table 1 for smoking status.
  • BMI category – Since height and weight data were collected online from participants by self-report, they are not measured. This should be explained in the section Methods
  • I don’t know if it is common to present whole survey, question by question, in results?? This refers to MD survey and AIS survey.

Discussion:

  • Discussion should start with a brief presentation of the main study results. You didn’t shortly present your results in this section.

Author Response

Reviewer Three Comments and feedback:

Dear Reviewer 

Thank you for your impressive feedback. Below you will find the points you raised and the responses written in red color.

Materials and methods:

  • How did you determine sample size? Based on specified inclusion criteria what was response rate?

Response: “The sample size was estimated for our survey to be around 385 participants [55] assuming a confidence level of 95% (95% CI) that the real value is within 5.0% (alpha level) and type II error of about 20% of the surveyed values of the female population in the Middle East and North Africa (223,874,467) according to the latest World Bank statistics [56]. In the final analysis, we planned to add around 917 individuals to raise statistical power to 90 % [55].”

  • How do you explain the difference in the number of participants by gender?

Response: we excluded the males from the study and now it is for Arabic-speaking female adults.

  • Ethical approval – Since it is multinational study, whether you should get ethical approval from all countries included in study? What happened to the participants who didn’t sign consent form?

Response: the participants were residents in Jordan with different nationalities, therefore, we did not need ethical approval from all countries. the participants who didn’t sign a consent form or wished to withdraw were excluded from the study

Results:

  • You said that about 90% of participants were female. That is not about two-third as you wrote, because two-third is about 66%. It is similar with male. Also, you should tune percentages in text and in Table 1 for smoking status.

Response: upon your feedback we changed the sample into females only. Taking into account that 90% of the study sample were females. Therefore, it was corrected as following:” The sample characteristics are shown in Table (1). The mean age of the sample was 36±10 years. The majority of the participants were single females (64%) and (86%) were non-smokers.”

  • BMI category – Since height and weight data were collected online from participants by self-report, they are not measured. This should be explained in the section Methods

Response: “Weight and height were obtained from the self-reported questionnaire and the BMI was generated accordingly.”

  • I don’t know if it is common to present whole survey, question by question, in results?? This refers to MD survey and AIS survey.

 Response: We thought that it was better to present all the questions to clarify the results. We can remove the detailed questions for better reading of the results.

Discussion:

  • Discussion should start with a brief presentation of the main study results. You didn’t shortly present your results in this section.

Response:In our study, 917 Arabic-speaking female adults with fourteen Arab nationalities who live in Jordan were included. Half of the participants were among the normal BMI (18.5 – 24.9 kg/m2). Nonetheless, the adherence of the MD diet was moderate amongst 56% of the sample. AIS was better among MD adherence score ≥10, with a mean increase of (6.02, 95% CI [5.28, 6.76]) and was statistically significant (p = 0.015), but no other group differences were statistically significant.”

Round 2

Reviewer 1 Report

Dear authors, 

Thank you for your changes and responses to my comments. The manuscript is much improved, although I would advise you to complete the tables in the paper further. 
Please find attached my minor comments. 

Kind regards, 

Author Response

Dear reviewer 1:

Thank you for your feedback concerning our manuscript (Manuscript id: 1463537). 

We want to take this opportunity to express our sincere thanks to you. Your efforts, experiences, and time are much appreciated.

We have studied the suggested corrections and your constructive comments carefully. We have responded to each of the comments raised by you. We hope that the manuscript is suitable for publication in its current form. Changes are highlighted in the red font; to facilitate the review process.

The following are the comments:

  1. Title: It might be good to add the word "women."

Response: Women was added to the title. "Arab Women Adherence to the Mediterranean Diet and Insomnia."

  1. Abstract:

Response: Below, reviewer one changes to the abstract were used as proposed.

"Background and objective: Sleeping difficulty affects overall health, nutrition, and wellbeing. The Mediterranean diet (MD) has proven its effect on improving the quality of life and overall health outcomes among people of all ages. Therefore, this study aims to determine whether adherence to an MD predicts reduced insomnia in Arabic-speaking female adults from fourteen Arab countries.

Methods: A self-administered, cross-sectional survey was used to assess the relationship between MD and sleep quality in Arabic-speaking female adults. Social media collected the data was collected from 917 Arabic-speaking women in Jordan between March and May 2021. All participants gave informed consent to participate and answered the whole questionnaire, including sociodemographic questions, MD score tool, Athens Insomnia Scale (AIS), and lifestyle components, including smoking and dietary patterns.

Results: The majority of the participants were Jordanian (57%) with an average age of 36 years. More than half of the participants reported moderate adherence to MD (6-9 points) (56%). In comparison, 36% and 13% reported low adherence (<6 points) and high adherence (≥10 points), respectively. According to AIS scores, most participants (59%) suffered from insomnia. One-way ANCOVA showed a statistically significant difference between MD adherence score categories and AIS, F (2, 914) = 3.36, p = 0.018. Women who reported high adherence to MD, compared with those who reported low adherence to MD, were less likely to suffer from insomnia assessed with the AIS (p = 0.015).

Conclusion: Our findings provide preliminary evidence that high adherence to MD may improve sleep and reduce insomnia in Arabic-speaking female adults. Further research is needed to confirm these results."

  1. Check throughout the manuscript that you use MD instead of Mediterranean Diet.

Response: checked and corrected

  1. Possibly, you should change the inclusion criterion "adult of both genders."

Response: the inclusion criteria was changed as follows:

"The participants were chosen from the general population based on the following inclusion criteria: (1) a female adult over the age of 18 (2) capable of writing and speaking Arabic (3) willing to give their information and volunteer to participate in the study. Participants who did not meet the inclusion criteria were excluded. Additionally, participants who did not consent or wish to withdraw were excluded from the study. "

  1. Line 158: It is correct "MD diet"? "Mediterranean diet diet?

Response: it was corrected, and the whole research was checked too.

  1. Tables 4, 5, 6 are not very informative. Most of them supposedly show information relating adherence to MD and insomnia measured with HAI. However, no data on both variables are presented in any of them. You should complete them. Here are some articles whose tables may inspire and help you.

https://pubmed.ncbi.nlm.nih.gov/33813674/ à DOI: 10.1007/s11325-021-02351-x (table 1)

https://pubmed.ncbi.nlm.nih.gov/33603907/àDOI:10.26574/maedica.2020.15.4.490 (table2)

Response: the tables were deleted, and the text reported the results.

  1. Strengths and limitations: I think you can emphasise the strength of having conducted a study on women. Possibly you can find some argument in the Gender-Sensitive Research standards.

Response: "Insomnia affects women more than men, even in the 21st century. Some women are simultaneously caring for children and aging parents. Women may also work outside the house while caring for others. These activities may affect women's sleep. First, hormonal changes in women's life phases. For example, women with the premenstrual dysphoric disorder (PMDD), a severe type of PMS, are more prone to sleeplessness. Then there's the fact that women become pregnant and have severe leg or stomach pains. Menopausal women often suffer from sleeplessness owing to hot flashes and excessive perspiration. Second, women are more prone to mood disorders like worry and sadness, making sleep issues more prevalent. Finally, women are more likely to be the primary caregivers."

Reviewer 3 Report

Materials and methods:

Study setting and participants

Please, indicate, based on specified inclusion criteria what was response rate?            

Perhaps you decided to exclude males from the study, you should change your first inclusion criteria.

Instrument

            The last sentence in the first paragraph of this section is repeated twice. 

Author Response

Dear reviewer 3:

Thank you for your feedback concerning our manuscript. Your time and efforts are much appreciated. The response was in a red-colored font.

Materials and methods:

Study setting and participants

Please, indicate, based on specified inclusion criteria what was the response rate?  

Response: response rate added.

" The participant's response rate to the online survey was (89.5%), and they answered all of the questions."         

Perhaps you decided to exclude males from the study, you should change your first inclusion criteria.

Response: first inclusion criteria changed.

"The participants were chosen from the general population based on the following inclusion criteria: (1) a female adult over the age of 18."

Instrument

            The last sentence in the first paragraph of this section is repeated twice. 

Response: the repeated sentence was deleted.